# Multilocus Phylogeography of the *Tuber mesentericum* Complex Unearths Three Highly Divergent Cryptic Species

**DOI:** 10.3390/jof7121090

**Published:** 2021-12-17

**Authors:** Marco Leonardi, Daniele Salvi, Mirco Iotti, Gian Luigi Rana, Aurelia Paz-Conde, Giovanni Pacioni

**Affiliations:** 1Department of Life, Health and Environmental Sciences, University of L’Aquila, Via Vetoio Loc. Coppito, 67100 L’Aquila, Italy; marco.leonardi@univaq.it (M.L.); daniele.salvi2@univaq.it (D.S.); giovanni.pacioni@univaq.it (G.P.); 2School of Agricultural, Forestry, Food and Environmental Sciences, University of Basilicata, Viale dell’Ateneo Lucano, 10, 85100 Potenza, Italy; gianluigirana@yahoo.com; 3Apto. de Correos 6, Caldes de Malavella, 17455 Girona, Spain; ita-paz@hotmail.com

**Keywords:** β-tubulin, EF1-α, ITS, truffles, typification, *Tuber suave* sp. nov., *Tuber bituminatum*

## Abstract

*Tuber mesentericum* is an edible European black truffle, apparently easy to recognize, but showing a high degree of genetic variability. In this study, we performed an integrative taxonomic assessment of the *T. mesentericum* complex, combining a multilocus phylogeographic approach with morphological analyses, and including authentic specimens of Vittadini, and Berkeley and Broome. We performed maximum likelihood phylogenetic analyses, based on single and concatenated gene datasets (ITS rDNA, β-tubulin, elongation factor 1-α), and including all available sequences from previous studies. Phylogenetic analyses consistently recovered three reciprocally monophyletic and well-supported clades: clade I, with a wide range across Europe; clade II, specimens collected mainly in the Iberian, Italian, and Balkan peninsulas; and clade III, specimens collected almost exclusively in central Italy. Genetic distance between clades ranged from 10.4% to 13.1% at the ITS region. We also designed new primer pairs specific for each phylogenetic lineage. Morphology of spores, asci, and peridium were investigated on specimens representing the three lineages. Macro- and micromorphological analyses of ascomata revealed only a few, but not diagnostic, differences between the three phylogenetic lineages, thus, confirming that they are morphologically cryptic. By studying authentic specimens of Vittadini, and Berkeley and Broome, it was possible to identify the three clades as *T. mesentericum*, *Tuber bituminatum*, and *Tuber suave* sp. nov., and to designate an epitype for *T. mesentericum* s.s. and a lectotype for *T. bituminatum*. Future investigations on volatile organic compound (VOC) composition are needed to define the aroma repertoires in this species complex.

## 1. Introduction

Truffles are hypogenous ascomata, mainly formed by fungi in the Ascomycota and Basidiomycota phyla. Those in the genus *Tuber* (Ascomycota, Pezizales), the so-called true truffles, live in mycorrhizal symbiosis with the roots of many trees, shrubs, and herbaceous plants [1]. *Tuber* species that produce ascomata with a pleasant smell and marketable size are considered edible and, therefore, of commercial interest. In some European countries, edible truffles have traditionally been considered one of the most appreciated and expensive foods, and their cultivation and consumption are now rapidly spreading worldwide [2].

Since the advent of molecular phylogenetics, many researchers have worked to revise the taxonomy within the genus *Tuber* [3,4,5,6]. Currently, about 120 *Tuber* species have been described and molecularly characterized from Asia, Europe, and Central and North America [7]. However, the last revision of the genus estimated from 180 to 220 species, grouped into 11 major clades [8]. A large portion of such species richness is due to the presence of cryptic lineages in many of these clades. Indeed, cryptic species have been found in *Tuber anniae* W. Colgan & Trappe [9], *Tuber borchii* Vittad. [10], *Tuber brumale* Vittad. [11], *Tuber excavatum* Vittad. [12], *Tuber indicum* Cooke & Massee [13,14], and *Tuber rufum* Picco [15]. However, the taxonomy of many of these species complexes has not been resolved.

The Aestivum clade contains the most morphologically diverse *Tuber* species [16]. In addition to truffles with a white (*Tuber magnatum* Picco) or brown (*Tuber panniferum* Tul. & C. Tul., *Tuber pulchrosporum* Konstantinidis, Tsampazis, Slavova, Nakkas, Polemis, Fryssouli & Zervakis) peridium, most species in the clade have black warty ascomata with alveolate spores (*Tuber aestivum* Vittad., *Tuber mesentericum* Vittad., *Tuber sinoaestivum* J.P. Zhang & P.G. Liu, *Tuber malençonii* Donadini, Riousset, G. Riousset & G. Chev.). Phylogenetic and phylogeographic studies in this clade have mainly focused on species with the highest economic values, such as *T. magnatum* and *T. aestivum* [17,18,19,20]. Within this clade, *T. mesentericum* is a truffle of some commercial interest, which is apparently easy to distinguish from other black truffles, based on its excavated base and a more or less strong bitumen-like smell, which has always been considered unpleasant by most authors [21].

Phylogenetic studies on *T. mesentericum* are scarce. Pacioni and Pomponi [22], based on allozyme polymorphisms, recognized four distinct clades within *T. mesentericum*, one of which was later associated with *Tuber bellonae* Quélet [23]. Sica et al. [24] analyzed rDNA internal transcribed spacer (ITS) sequences in 126 specimens, mainly from the Campania region (southern Italy), and found 34 distinct haplotypes, but did not reveal any significant morphological differences among the analyzed ascomata. Benucci et al. [25] and Marozzi et al. [26] uncovered three distinct clades within *T. mesentericum*, using ITS and elongation factor 1-α (EF1α) sequences, respectively. These lineages may well represent cryptic species, one of which is *T. mesentericum* sensu stricto. A proper taxonomic assessment of the *T. mesentericum* species complex has not been performed yet, and it is unclear whether phylogenetic lineages are also distinct by any morphological feature, or by what their geographic distributions are.

In this study, we performed a comprehensive phylogeographic assessment of the *T. mesentericum* species complex, based on multilocus data, and including specimens from all over Europe. The taxonomy of the lineages found was resolved through an integrative approach combining molecular data, morphological characters of peridium and spores, and analysis of voucher specimens of *T. mesentericum* determined by Vittadini, and the type specimens of *T. bituminatum* determined by Berkeley and Broome. The new species *Tuber suave* sp. nov. was also attributed to one of these lineages.

## 2. Materials and Methods

### 2.1. Specimens

A total of 51 voucher specimens identified as *T. mesentericum* and deposited in the herbaria of L’Aquila (AQUI), Eötvös Loránd (BPU), Uppsala (UPS), Duke (DUKE), Bologna (CMI-UNIBO) Universities, and the Civic Museum of Venice (MCVE) were sequenced and used in phylogenetic analyses (Table 1). Analyzed specimens also included one of Vittadini’s authentic specimens (K(M)190347), and three specimens designed as holotypes of *T. bituminatum* (K(M)30594, from the herbarium of C. E. Broome), preserved at the Herbarium of the Royal Botanic Gardens, Kew (Appendix A, Table 1 and Appendix A).

### 2.2. Molecular Analyses

#### 2.2.1. DNA Extraction, Amplification, and Sequencing of Voucher Specimens

Each sample was ground in a TissueLyser II (Qiagen, Hilden, Germany) for 1 min at 30 Hz. Genomic DNA was isolated from ~20 mg of dried specimen, using the DNeasy Plant Mini Kit (Qiagen, Hilden, Germany), following the manufacturing protocol. DNA quality and concentration were assessed using a NanoDrop ND-1000 Spectrophotometer (Thermo Fisher Scientific, Madison, WI, USA). ITS regions, the β-tubulin (β-tub) gene, and elongation factor 1-α (EF-1-α) were amplified by polymerase chain reaction (PCR) for multigene analysis. Amplifications were performed using the primers ITS1F [27] and ITS4 [28], Bt2a/BT2b [29], and EFtFw/EFtBw [30] (primers specific for Ascomycetes), respectively. Amplifications were performed in 50 µL volume reactions containing 25 µL of BioMix^TM^ (2×) (Bioline Reagents Ltd., London, UK), 200 nM of each primer, 10 µg of BSA (Bovine Serum Albumine, Roche Diagnostics GmbH, Mannheim, Germany), and ~10 µg of extracted DNA. The PCRs of the ITS regions were run with the following cycling protocol: initial denaturation at 94 °C for 2 min and 30 s; 25–30 cycles at 94 °C for 30 s, 55 °C for 30 s, and 72 °C for 45 s; and final extension at 72 °C for 7 min. Thermal protocols for amplification of β-tub and EF-1-α are reported in Paolocci et al. [30]. PCR products were purified using the QIAquick PCR Purification Kit (Qiagen, Milan, Italy) and sequenced by Eurofins Genomics Service (Ebersberg, Germany). Sequence reactions were carried out in both directions with the primer pairs ITS1F–ITS4, tubtubf/elytubr [31] (primers specific for β-tub of *Tuber* spp.), and EFtFw/EFtBw. Sequences were edited with BioEdit [32] and compared with those present in the GenBank (https://www.ncbi.nlm.nih.gov/BLAST/, accessed on 2 July 2021) database, using the BLASTn search [33]. Sequences generated in this study were deposited in the GenBank with the accession numbers: KP686239–KP686246, KT456261–KP456267, KX290123–KX290124, OL711588–OL711628, OL753348–OL753432 (Table 1).

#### 2.2.2. ITS Amplification and Sequencing of Historical Voucher Specimens

Due to the low amount and bad quality of fungal material, the four historical voucher specimens from the Kew herbarium were amplified by a direct PCR approach, using newly designed primer pairs specific for each cryptic lineage, inferred from phylogenetic analyses.

Because of the high degree of DNA degradation of the historical voucher specimens, the lineage-specific primer pairs were targeted to short segments (under 380 bp) of the ITS regions. To this aim, all ITS sequences of the *T. mesentericum* complex considered in this study were aligned with BioEdit, and the most informative domains of ITS1 and ITS2 regions were identified to design forward and reverse primers, respectively (Appendix A). Primer selection was performed using Primer Express 3.0 (PE Applied Biosystems, Waltham, MA, USA) and Primer3 v 4.1.0 [34]. The specificity of each primer pair was preliminarily tested on in silico analyses, and then on DNAs extracted from ascomata of *Tuber* species genetically closely related to *T. mesentericum* (*T. melanosporum*, *T. brumale*, *T. aestivum*, *T. indicum*, and *T. magnatum*). The sequences of the lineage-specific primers designed in this study are reported in Table 2.

Small fragments of dry tissue (<1 mg) were transferred to 0.2 mL PCR tubes and rehydrated with 7 µL of ultrapure water, and then incubated at 65 °C for about 30 min before being submitted to a freeze–thaw process (1 min in liquid nitrogen followed by 1 min at 65 °C, three times) to facilitate tissue disruption and DNA release. Sample lysis was enhanced by using sterile glass micropestles specifically tailored for 0.2 mL tubes. The PCRs were carried out in 50 µL volume reactions containing 25 µL of BioMix^TM^ (2×), 200 nM of each primer, and 40 µg of BSA. The PCRs for each ITS region were run with the following cycling protocol: initial denaturation at 95 °C for 6 min; 40 cycles at 94 °C for 45 s, 53 °C for 30 s, and 72 °C for 45 s; and final extension at 72 °C for 7 min. The PCR products were purified and sequenced, as described above.

#### 2.2.3. Phylogenetic Analyses

DNA sequences were checked and assembled using Geneious R8 (Biomatters Ltd., Auckland, New Zealand). *Tuber melanosporum* and *T. magnatum* were used as outgroup taxa (Appendix A). Multiple sequence alignments for the three loci were performed in MAFFT7 [35], using the E-INS-i iterative refinement algorithm.

Maximum likelihood (ML) phylogenetic analyses based on single-gene datasets (β-tub, EF-1-α, and ITS), as well as on the concatenated dataset β-tub + EF-1-α + ITS, were performed. The ITS dataset was significantly enlarged by adding all available sequences from previous studies (mainly from Sica et al. [24]) to maximize geographic coverage of samples. Maximum likelihood analyses were performed in raxmlGUI 1.5b2 [36], a graphical front end for RAxML 8.2.1 [37], with 100 independent ML searches, 100 bootstrap replicates, and 1000 rapid bootstrap replicates, and by applying the best-fit partitioning scheme and substitution models selected by PartitionFinder 2 [38]. For the concatenated dataset, a partitioned model with four partitions received the highest score under Akaike’s information criterion (ITS1 and ITS2: GTR + I + G; 5.8S: GTR + I + G; β-tub GTR + G; EF-1-α GTR + I + G).

### 2.3. Morphological Analyses

Microscopic characters of spores, asci, and peridium were examined on hand-made sections or squash preparations obtained from 16 specimen vouchers (5–6 for each cryptic lineage, Table 1). Each sample was rehydrated for 10 min in 20% KOH, rinsed with sterile water, and then soaked with 3% KOH, following the procedure described by Leonardi et al. [39]. Observations and measurements were made under a Zeiss AXIO imager2 microscope and images were captured by a Leica DFC320 camera.

Only fully mature spores in which the episporia were clearly distinguishable were considered for the analyses. The color of the spores was determined using the Rayner [40] (R) mycological color chart and the ColorHexa (H) color scale (https://www.colorhexa.com/color-names or https://www.w3schools.com/colors/colors_picker.asp, accessed on 10 November 2021), at 400× magnification with a 5000° K light source, without a filter. The measurements of the microscopic characters were carried out at 400× or 1000× magnification. The following spore characters were measured (Table 3, Appendix A): L1 and W1, and spore length and width, excluding episporium (cell lumen); L2 and W2, and spore length and width, including episporium; L3 and W3, and spore length and width, including episporium and exosporium (ornamentations); Q, length/width ratio (L1/W1, L2/W2, L3/W3); episporium thickness (spore wall); and exosporium thickness (height of ornamentations). L2 and W2 were also measured from one-, two-, three-, four-, five-, and six-spored asci, when present. In addition, dimensions of the asci, peridial elements (layers and cells of the exo- and endoperidium), and warts were also measured. Measurements were reported in the text as minimum and maximum values, while means ± standard deviations are reported in Table 3.

Differences in spore size (L2, W2, and L2/W2) among the cryptic lineages were evaluated by one-way repeated measure ANOVA, with repeated measures using the AoV function (measure~species + Error(ascoma)) in R [41]. Statistical analyses were carried out using log-transformed data to account for the effect of data skewness on the statistical tests.

## 3. Results

### 3.1. Phylogenetic Analyses

Phylogenetic analyses based on either single genes or on the concatenated dataset consistently recovered three reciprocally monophyletic clades (Figure 1 and Figure 2) that received maximum support (bootstrap support = 100) in the concatenated analysis: (i) one clade with the widest range across Europe (clade I); (ii) a second clade grouping specimens collected mainly in the Iberian, Italian, and Balkan peninsulas (clade II); and (iii) a third clade including specimens collected almost exclusively in central Italy and Greece (clade III). The ranges of the three clades were found to overlap in different European regions, particularly in western Europe, Italy, and Greece (Figure 3).

Genetic divergence among the three clades ranged from 0.097 to 0.123 (uncorrected *p*-distance) and from 0.104 to 0.131 (Kimura-2-parameter distance) at the ITS region; from 0.037 to 0.053 (uncorrected *p*-distance) and from 0.034 to 0.054 (Kimura-2-parameter distance) at the β-tubulin region; and from 0.026 to 0.034 (uncorrected *p*-distance) and from 0.027 to 0.032 (Kimura-2-parameter distance) at the EF-1-α region. Intra-clade divergence at the ITS region was 0.0047 for clade I, 0.0074 for clade II, and 0.0032 for clade III, using either *p*-distance or Kimura-2-parameter estimates.

The ITS sequence of *T. mesentericum* Vittadini’s authentic specimen and that of one specimen of *T. bituminatum* from the Kew herbarium were clustered into clade I (Figure 1 and Figure 2). ITS sequences of the other two *T. bituminatum* specimens were grouped with samples of clade II (Figure 1 and Figure 2). All of these sequences were generated with the species-specific primer pairs designed in this study (Table 2). The efficiency of these specific primers was also tested on the specimens used for phylogenetic analyses, and on the other 21 additional specimens deposited in the AQUI and MCVE herbaria (Table 1 and Appendix A).

### 3.2. Ascoma Morphology

Micro- and macromorphological characters of ascomata showed a wide overlap among ascomata of different clades and have not proved useful for their taxonomic identification. Few differences were found, only in the shape of peridial cells constituting the center of warts and in the morphology of spore reticulum. Ascomata in clades I and II showed a pseudoparenchymatous peridium with globose–angular cells, whereas those in clade III showed a pseudoparenchyma composed of elongated cells that radiated out from the base of the warts (Figure 4C, Figure 5B and Figure 6B). The spores in clades I and III always showed regular and completely formed alveolae (Figure 4D and Figure 6C,D), whereas a variable percentage of those in clade II had labyrinthine pseudoreticula with incomplete and irregular alveolae (Figure 5C,D).

No statistical differences were found in any of the measured characters of the gleba hyphae, asci, or spores of the three clades. Nearly significant differences were found only for L2 (*p* = 0.051) and L2/W2 (*p* = 0.062) spore parameters when values were log-transformed before ANOVA (Appendix A).

## 4. Discussion

The phylogeny of the *T. mesentericum* complex has been poorly investigated to date, and only recently has it attracted the interest of mycologists. So far, three studies have investigated the genetic diversity of *T. mesentericum* by applying a single-gene analysis [24,25,26], but none of them have resolved the taxonomic issues within this species complex. In this study, we performed an integrative taxonomic assessment of the *T. mesentericum* complex by combining a multilocus phylogeographic approach with in-depth morphological analyses, and including authentic specimens of Vittadini and Berkeley and Broome. This approach allowed us to resolve the main phylogenetic clades found within the *T. mesentericum* complex into three species and to designate an epitype for *T. mesentericum* s.s.

In agreement with previous studies by Benucci et al. [25] and Marozzi et al. [26], we found three distinct and well-supported lineages, both in single-locus and multi-locus phylogenetic analyses.

Phylogeographic data generated in this study combined with data from public repositories indicate that members of clade I are spread all over Europe, from Sweden to Italy and from Spain to Bulgaria, and are associated with many hardwood trees, including *Quercus* spp. (*Q. cerris* L., *Q. robur* L.), *Fagus sylvatica* L., *Corylus avellana* L., *Ostrya carpinifolia* Scop., but also with conifers, such as *Picea* spp. The range of clade II seems to be mainly limited to Mediterranean habitats, and their members show a similar host preference. Clade III is only distributed in central Italy and Greece, although a specimen from GenBank (JQ348414) had “France” as its general indication of origin, which should be verified. To date, ascomata of Clade III have been found only under thermophilic oaks.

Clades I and II seem to be sister species that diverged after clade III. While the average intraclade divergence at the ITS marker was <0.8%, the divergence between these clades was as high as 10% or 13%, values that are well above those observed among the currently accepted *Tuber* species. For example, interspecific divergence observed between closely related pairs of currently recognized species belonging to Rufum and Melanosporum clades are in the range of 4.7–7.4% [39]. Bonito et al. [42] suggested a threshold for *Tuber* species delimitation based on an interspecific divergence of ≥4%. However, interspecific divergence below 4% has been reported between closely related species of different clades, such as the Excavatum [12], Puberulum [9], and Gibbosum [43] clades.

Morphological and biometric characters of ascomata examined in this study do not allow an unambiguous identification of the clade to which they belong, thus confirming that these lineages represent cryptic species. Only two of the microscopic characters, the peridium (cell pattern of the peridium cells in the center of the single warts) and spores (reticulum integrity), showed some differences among clades, although they are not diagnostic and, therefore, of little taxonomic utility. ANOVA analyses based on spore sizes showed low differentiation among clades, with only spore length (L2) being marginally significant.

The mismatch between phylogenetic and morphological differentiation seems to be a common feature in the genus *Tuber*. Bonuso et al. [10] identified two genetically isolated groups in *T. borchii*, but no distinctive morphological features were found in support of this separation. Ascoma morphology has not been useful in solving the taxonomy of the *T. indicum* complex, where at least two cryptic species exist, according to different phylogenetic studies [13,14,44]. The morphological distinction between *T. brumale* and *T. cryptobrumale* is also challenging; Merényi et al. [11] verified that only a combination of different morphological characters would make it possible to differentiate most (95%) specimens belonging to these two pseudocryptic species. The issue of cryptic species seems even more complex in other *Tuber* lineages. Five different species complexes were identified by Healy et al. [15] in the Rufum clade, and as many cryptic species were found by Puliga et al. [12] for the European truffle *T. excavatum*. From a taxonomic point of view, the greatest challenge will be to define and name each cryptic species within these complexes.

The integrative approach used in this study has revealed itself to be particularly fruitful in resolving the taxonomy of the cryptic species complex of *T. mesentericum*. The genotyping of authentic specimens of Vittadini, and Berkeley and Broome from the Kew herbarium allowed us to identify clade I as *T. mesentericum* sensu stricto, clade II as *T. bituminatum*, and clade III as a new species named *T. suave*, for the pleasant smell that seems to be typical of its ascomata. The type of *T. mesentericum*, as well as the types of the other *Tuber* species described by Vittadini [45], are no longer available for genotyping. Vittadini’s private herbarium was never deposited in a public institution, and it was lost because of insect infestation [46]. Only a few authentic specimens determined as *T. mesentericum* by Vittadini are actually available at the herbaria of Kew, Padua, and Uppsala (Figure 4B and Appendix A). The specimen labelled as *T. mesentericum* (K(M)190347), and one of the three specimens labelled as *T. bituminatum* from the Kew herbarium (Figure 4B and Appendix A), were successfully amplified by using the ITS primer pair newly designed for the members of clade I. On the contrary, the PCR amplifications of the specimens deposited in the Padua and Uppsala herbaria (Appendix A) were unsuccessful, regardless of the lineage-specific primer pair designed in this study. This was probably due to their poor state of preservation, biological contamination, or the chemicals added to preserve the exsiccata. The iconography of Vittadini [45] (Table III, Figure XIX) constitutes an ambiguous lectotype because it only represents a detail of the gleba, and not the entire ascoma with the warty black peridium and the basal cavity, hence the epithet “mesentericum”. Therefore, we designated a specimen collected in Vittadini’s study area as an epitype of *T. mesentericum* s.s. This specimen has an ITS sequence identical (100% identity) to that of the authentic specimen voucher from the Kew herbarium (K(M)190347) (Figure 4B).

*Tuber bituminatum* was described by Berkeley and Broome [47], but was later considered a synonym of *T. mesentericum* or *T. aestivum* [48,49,50]. Our analyses demonstrated that two out of three specimens indicated by Berkeley and Broome as holotypes of *T. bituminatum* fall into clade II and, therefore, this species should be revaluated and exclusively attributed to the members of this lineage within the *T. mesentericum* complex. Recently, Crous et al. [51] described the new species *Tuber alcaracense* Ant. Rodr. & Morte, having ascomata similar to those of species of the *T. mesentericum* complex, but with a pleasant odour and lacking a basal cavity. A preliminary phylogenetic comparison of the two ITS sequences available for *T. alcaracense* (MN810046-7) with the ITS sequences generated in this study indicates that *T. alcaracense* is nested within the clade of *T. bituminatum* (clade II, results not shown). Therefore, most likely *T. alcaracense* is a synonym of *T. bituminatum*, as the latter name has priority. However, before drawing taxonomic conclusions in this regard, more in-depth morphological and phylogenetic analyses must be carried out.

As we observed for the Kew herbarium, it is possible that many collections labelled as *T. mesentericum* in private or public herbaria can also include *T. bituminatum* and/or *T. suave* specimens, due to the scarcity of morphological traits useful in distinguishing them. For example, the authentic specimen of *T. mesentericum* preserved in the PD herbarium (Milano 1841, leg. et det. C. Vittadini) for its labyrinthine spore ornamentation might belong to *T. bituminatum*.

The aroma of ascomata within the *T. mesentericum* complex still remains controversial. In our analysis, *T. mesentericum* s.s. generally had a smell of solvent, whereas the smell of *T. suave* ascomata was consistently pleasant, as also confirmed by Marozzi et al. [26] for the specimens of the same lineage. On the contrary, the smell of *T. bituminatum* was very variable: it was often unpleasant, mainly due to the production of 1-methoxy-3-methylbenzene (reported as 3-methyl-anisole by [52]), and sometimes it was very pleasant. Although studies on the volatile organic compound (VOC) composition of *T. mesentericum* have been carried out in the past [53], an extensive investigation on ascomata of this species complex should be planned in light of recent taxonomic insights.

## 5. Taxonomy

### 5.1. Tuber mesentericum Vittad.

MB# 180551

**Lectotype**: Vittadini C. 1831, Monographia Tuberacearum, Table III Figure XIX

**Epitype**: Italy, Lombardia, Parco Reale di Monza, 45°35′41″ N 9°16′26″ E, 175 m asl, *O. carpinifolia*, 5 October 2017, leg. Stefano Seghezzi, det. Giovanni Pacioni (AQUI 9717), (Figure 4A,C,D); MTB# 10004606.

GenBank: OL711593 (nrITS), OL753399 (nr β-tub), OL753354 (nr EF1-α).

*Ascomata* hypogeous, globose to slightly lobed, more or less hollowed at the base, 2–7 cm in diameter. Peridium blackish, covered with flat pyramidal warts. Warts 3–6 sided, up to 4 mm wide, 1.5–2.0 mm high, bearing 3–5 vertical grooves near the middle of the face, sometimes with rough and fine horizontal striae, apex open and depressed. *Odour* pleasant, sometimes with a faint solvent (similar to trichloroethylene) component, disappearing when the context is exposed. *Peridium* pseudoparenchymatous, 150–250 µm; outer layer (exoperidium) 50–90 µm thick, almost opaque, composed of angular, irregular cells 10–20 µm in diameter, with somewhat thickened (4–5 µm), dark red-brown walls and reduced cell lumen (“sclerenchymatous” type); inner layer (endoperidium) hyaline or pale yellow, 90–160 µm thick, composed of densely packed globose-angular cells, 8–28 × 8–20 µm in size, with thickened walls (2–4 µm) (“collenchymatous” type), merging with glebal tissue of interwoven hyphae. *Gleba* brown (R: from “88-hazel” to “63-sepia”; H: “#ad5c33-dark orange” or “#8b0000-dark red”, from “#5f2f00” to “#472300-very dark orange”), with whitish sterile veins, meandering or radiating from the cavity or base, composed of 3–6 µm wide, hyaline and interwoven hyphae.

*Asci* clavate–subglobose 1–6 spored, 80–100 µm, sometimes thick-walled when ripe, with a stalk 4–18 µm long, 6–9 µm wide, arising from a crozier 9–13 µm wide. *Ascospores* regularly reticulated, ellipsoid, rarely globose or elongated (Q2 = 1.00–1.69), straw (R: 46), pure yellow (R: 14), or yellow-orange (R: from 11-pale luteous to 12-luteous; H: from “#ffe066-light yellow” to “#ffae42-yellow orange”); spore lumen (L1 × W1) = 18–53 × 16–31 µm, episporium (L2 × W2) = 24–61 × 20–41 µm, including exosporium (L3 × W3) = 30–70 × 26–56 µm, spore wall thickness (episporium) 1.5–5.0 µm. Ascospores (L2 × W2) in 1-spored asci 33–61 × 22–48 µm, 2-spored asci 22–44 × 22–33 µm, 3-spored asci 23–38 × 18–30 µm, 4-spored asci 21–39 × 19–26 µm, 5-spored asci 23–34 × 20–28 µm, 6-spored asci 24–32 × 21–26. Exosporium consisting of a narrow to a wide reticulum of complete polygonal alveolae with 3–7 sides, 2–8 µm high, 1.5–10 µm wide; meshes from 2–4 to 7–11 and from 2–3 to 5–9 across spore length and width, respectively.

Habitat: beneath ectomycorrhizal hardwoods and conifers, especially beech on calcareous soils, mainly in autumn.

Additional specimens examined: **Bulgaria:** Vratsa, Chelopek, *Carpinus orientalis* Mill., 1 August 2017, leg. T. Nedelin, det. G. Pacioni (AQUI 10144); Loveč, Letnitsa, *Carpinus betulus* L., 16 August 2018, leg. T. Nedelin, det. G. Pacioni (AQUI 10233); **France**: Occitanie, Castelnau-de-Montmiral (Tarn), *Pinus sylvestris* L. and *C. avellana*, 15 October 2019, leg. L. Gerola (AQUI 10234); Roussayrolles (Tarn), *Quercus pubescens* Willd. and *C. avellana*, 15 October 2019, leg. L. Gerola (AQUI 10235); Milhars (Tarn), Bonan, *Q. pubescens*, *C. avellana*, and *C.*
*betulus*, 15 October 2019, leg. L. Gerola (AQUI 10238, AQUI 10239); Bruniquel (Tarn-Garonne), *Q*. *pubescens*, *C**. avella**na*, and *C**. betulus*, 16 October 2019, leg. L. Gerola (AQUI 10236); Le Montat (Lot), *C**. avellana*, 16 October 2019, leg. L. Gerola (AQUI 10237); **Great Britain**: Wiltshire, Bowood, holotype of *T. bituminatum* Berk. and Broome (K(M)30594, specimen 01); Bowood, *Q. robur* and *F. sylvatica*, 27 March 2018, leg Y. Borukov, det. G. Pacioni (AQUI 10140, AQUI 10141); **Greece**: W. Macedonia, Kastoria, *Populus* sp., *C**. avella**na*, and *C**. betulus*, 5 October 2013, leg. P. Kladopoulou, det. G. Pacioni (AQUI 10230, AQUI 10243); W. Thrace, Xanthi, *F. sylvatica*, 11 November 2007, leg. P. Kladopoulou, det. G. Pacioni (AQUI 6558); **Hungary**: Heves, Felsőtárkány, Mt. **Bükk****, *F. sylvatica*, October 2000, leg. I. Kiss, det. Z. Bratek** (ZB 2077-BPU, ZB 2092-BPU, ZB 2228-BPU); **Italy**: “*Vittadini misit*” (K(M)190347); Abruzzo, L’Aquila (AQ), Vasto di Assergi, *O. carpinifolia*, 12 November 1988, det. G Pacioni (AQUI 6548); Lucoli (AQ), Campo Felice, *F. sylvatica*, 25 November 2012, leg. D. Marinucci, det. G. Pacioni (AQUI 8510, AQUI 8511); Roccaraso (AQ), Pietransieri, *Q.cerris*, and *F. sylvatica*, 11 November 2012, leg. P. Oddis, det. G. Pacioni (AQUI 7227); Collelongo (AQ), Prati di S. Elia, *F. sylvatica*, 21 November 2014, leg. M. Cassetta, det. G. Pacioni (AQUI 8990); Calabria, Morano Calabro (CS), Campiglioni, Bosco Donna Calda, 9 October 2019, det. A. Paz-Conde (AQUI 10326/IC9101901); Campania, Bagnoli Irpino (AV), 3 November 2011, det. G. Pacioni (AQUI 10324); Friuli-Venezia Giulia, Maniago (PN), *O. carpinifolia*, 16 June 2002, leg. and det. G. Zecchin (MCVE 24188); Montereale Valcellina (PN), 25 October 2009, leg. A. Dal Cin, det. E. Campo (MCVE 25329); Lombardia, Bellagio (CO), 3 October 2017, leg. A. Bincoletto, det. G. Pacioni (AQUI 10145); Lecco, 4 September 2017, leg. S. Seghezzi, det. G. Pacioni (AQUI 9716); **Spain**: Catalonia, Maçanet de Cabrenys (Girona), *F. sylvatica*, *Quercus* sp., and *Picea* sp., 13 March 2017, leg. A. Paz and C. Lavoise, (AQUI 9718/IC13031703); Cantabria, Tudanca, Reserva Nacional Nansa, *F. sylvatica* and *C. avellana* L., 28 September 2015, leg. A. Paz and C. Lavoise (AQUI 10138/IC28091526); **Sweden**: Gotland, Tingstäde, *Q. robur* and *C. avellana*, 7 October 2000, leg. C. Wedén (UPS F-118824-CW005/DUKE 0348880).

The ITS and EF1-α sequences of this species fall into clade I after Benucci et al. [25] and into subclade I, after Marozzi et al. [26], respectively.

### 5.2. Tuber bituminatum Berk. & Broome

MB# 196523

**Lectotype**: Great Britain, Whiltshire, Bowood, October 1847, (K(M)30594) specimens 02 and 03 (Figure 5A–D); MTB# 10004607

GenBank: OL711615 and OL711616 (nr ITS).

*Ascomata* hypogeous, globose, more or less hollowed at the base, 2–10 cm in diameter. *Peridium* blackish with flat pyramidal warts. Warts 4–6 sided, up to 3 mm wide, 1–1.5 mm high, with flat or center grooved faces, with rough horizontal striae, apex pointed, sometimes depressed. *Odour* from phenolic (hence the epithet “*bituminatum*”) to very pleasant. *Peridium* pseudoparenchymatous; exoperidium 80–120 µm thick, almost opaque, composed of angular, irregular cells 8–14 µm in diameter, with somewhat thickened (4–5 µm), dark-brown walls and reduced cell lumen (“sclerenchymatous” type); endoperidium hyaline or pale yellow, 100–150 µm thick, composed of tightly packed globose–angular cells, 11–32 × 8–14 µm in size, with thickened walls (up to 5 µm) (“collenchymatous” type), merging with glebal tissue of interwoven hyphae. *Gleba* brown (R: from “88-hazel” to “63-sepia”; H: from “#84329-dark red” to “#472300-very dark orange”), with whitish sterile veins, meandering or radiating from the cavity, composed of 2–6 µm wide, hyaline and interwoven hyphae.

*Asci* clavate–subglobose, 1–6 spored, 70–100 µm, sometimes thick-walled when ripe, with a stalk 8–30 µm, arising from a crozier 9–14 µm wide. *Ascospores* regularly and irregularly (“labyrinthine”) reticulated, ellipsoid, sometimes globose or fusiform (Q = 1.00–2.09), yellow (R: “14-pure yellow”; H: from “#ffd633-vivid yellow” to “#ffdb4d-light yellow”) or orange (R: from “12-luteous” to “47-amber”); cell lumen (L1 × W1) = 16–52 × 14–33 µm, episporium (L2 × W2) = 22–61 × 20–43 µm, including exosporium (L3 × W3) = 29–70 × 26–54 µm, spore wall thickness (episporium) 1.5–6.5 µm. Ascospores (L2 × W2) in 1-spored asci 36–61 × 30–38.5 µm, 2-spored asci 28–43 × 23–35 µm, 3-spored asci 25–38 × 22–32 µm, 4-spored asci 22–34 × 19–29 µm, 5-spored asci 27–34 × 22–26 µm, 6-spored asci 28–32 × 22–28 µm. Exosporium consisting of a mixture of 3–5 sided, complete and irregular alveolae whose main ridges are broken, branched, and interconnected with minor ridges, 2.0–9.5 µm high, 2.0–10 µm wide; meshes 4–5 and 3–4 across spore length and width, respectively.

Habitat: beneath ectomycorrhizal hardwoods, but also conifers, especially beech on calcareous soils, mainly in autumn.

Additional specimens examined: **France**: Aquitaine-Limousin-Poitou-Charentes, La Gripperie-Saint-Symphorien (Charente-Maritime), *F. sylvatica*, *Quercus* sp., and *Picea* sp., 14 November 2014, leg. P. Chautrand (AQUI 10136/IC14111417); **Greece**: Peloponnese, Messene, Mt. Taygetos, *Q. pubescens*, 10 March 2013, leg. P. Kladopoulou, det. G. Pacioni (AQUI 8579); **Italy**: Abruzzo, Lucoli (AQ), Campo Felice, *F. sylvatica*, 19 October 2010, D. Marinucci (AQUI 7097); Lucoli (AQ), Campo Felice, *F. sylvatica*, 8 November 2012, leg. D. Marinucci, det. G. Pacioni (AQUI 8417); Lucoli (AQ), Prato Capito, *F. sylvatica*, 27 November 2011 (AQUI 7162); L’Aquila (AQ), Arischia, *Q. pubescens*, 1 November 1976 (AQUI 3285); L’Aquila (AQ), Vasto di Assergi, *C. avellana* and *O. carpinifolia*, 13 September 1988 (AQUI 6505, AQUI 6509); Cappadocia (AQ), Camporotondo, *F. sylvatica*, 18 December 2016, leg. I. Apolloni, det. G. Pacioni (AQUI 9833); S. Vincenzo Valle Roveto (AQ), Morrea, *F. sylvatica*, 15 December 2014, leg. M. Cassetta, det. G. Pacioni (AQUI 9713); Gamberale (CH), Laghetto, *F. sylvatica*, 28 November 2016 (AQUI 9707); Palena (CH), Madonna dell’Altare, *F. sylvatica* and *Q. cerris*, 25 November 2016 (AQUI 9702); Palena (CH), Arsiccia, *F. sylvatica* and *Q. cerris*, 2 December 2016 (AQUI 9703); Palena (CH), Val di Terra, *F. sylvatica*, 4 December 2016 (AQUI 9704); Basilicata, Abriola (PT), Mt. Pierfaone, *F. sylvatica*, 11 November 2011, leg. and det. G.L. Rana (AQUI 8994, AQUI 8995); Pignola, La Sellata-Rifreddo, *F. sylvatica*, 11 November 2011, leg. and det. G.L. Rana (AQUI 8991, AQUI 8992, AQUI 8993); Calabria, Rossano Calabro (CS), Mediterranean maquis, 12 December 2000, leg. Di Leone (MCVE 16209); Morano Calabro (CS), Campiglioni, Bosco Donna Calda, 9 October 2019, leg. A. Paz-Conde (AQUI 10325, AQUI 10327); Campania, Colliano (SA), *F. sylvatica*, 10 October 2001, leg. and det. A. Zambonelli (CMI-UNIBO 1807); Umbria, Perugia (AQUI 8921) and central Apennines without indications (AQUI 7099; AQUI 8919; AQUI 8920); **Spain**: Cantabria, Comillas, Reserva Nacional Saja, *F. sylvatica*, 27 February 2015, leg. A. Paz and C. Lavoise (AQUI 10134/IC27021515); Cabezon de la Sal, Mt. Corona, *Abies alba* Mill., 15 October 2016, leg. A. Paz and C. Lavoise (AQUI 10137/IC15101604); Valdaliga, Bustriguado, *A. alba*, 15 October 2016, A. Paz and C. Lavoise (AQUI 10139/IC15101621).

The ITS and EF1-α sequences of this species fall into clade II after Benucci et al. [25] and into subclade II after Marozzi et al. [26], respectively.

### 5.3. Tuber suave Pacioni & M. Leonardi, sp. nov.

MB# 842206

**Holotype**: Italy, Abruzzo, L’Aquila, Secinaro, *Q. pubescens*, leg. Antonio Barbati, 11 November 2010 (AQUI 7131) (Figure 6A–D)

GenBank: OL711623 (nr ITS), OL753414 (β-tub), OL753375 (nr EF 1-α).

Etymology: from the Latin ‘*suave*’ (neuter), pleasant for its flavor.

Diagnosis: Ascoma with black and flat pyramidal warts, and flat or excavate base; peridium pseudoparenchymatous textura porrecta with parallelepiped cells that decrease in size from the inside out, while increasing the thickness and pigmentation of the walls. Ascospores yellowish, ellipsoid 22–60 × 18–40 µm without ornamentations, exosporium 2–7 µm high, alveolate with complete polygonal meshes with 4–7 sides. Smell and taste pleasant. Under thermophilic oaks in the Mediterranean area.

*Ascomata* hypogeous, globose, more or less hollowed at the base, 4–5 cm in diameter. Peridium black, covered with flat pyramidal warts. Warts 4–6 sided, up to 2–3 mm wide, 1.5–2 mm high, grooved faces, without horizontal striae, apex pointed or open, and depressed. *Odour* complex, intense, and pleasant. *Peridium* pseudoparenchymatous, 100–250 μm thick; exoperidium 40–100 μm thick, opaque, composed of elongated cells (outermost layers), with deep-brown walls (R: from “9-umber” to “63-Sepia”; H: from “#663300–nutmeg wood finish” to “#661a00–red oxide”), strongly thickened (3–5 μm) and reduced or absent cell lumen (“sclerenchymatous” type), 6–17 × 4.5–8 μm, followed by a palisade layer (55–100 μm thick) of elongated cells completely filling the wart, 12–40 × 6–14 μm, with the major axis perpendicular to the surface; endoperidium 50–150 μm thick, yellowish (R: “45–buff”; H “#ff6600–safety orange”, “#e65c00–persimon”, “#e287A3–burnt Sienna”), composed of a mixture of pseudoparenchymatous cells, 4–11 × 2–7 μm, with 2–3 μm thickened walls (“collenchymatous” type). *Gleba* deep brown (R: “63-Sepia”; H: “#5f2f00-baker’s chocolate” or “472300-very dark orange”), with white, meandering, rather wide sterile veins, composed of interwoven, thin-walled, 3–5 µm wide hyphae. *Asci* 1–5 spored, clavate–suglobose, 55–100 μm, sometimes thick-walled, with a stalk up to 30 µm long, 6–8 μm wide, arising from a crozier 9–12 μm wide.

*Ascopores* reticulated, ellipsoid, sometimes globose or slightly elongated (Q = 1.00–1.75); yellowish or burnt orange (R: from “47-amber” to “9-umber”; H “#e6b800-pure yellow”, from “#ffd119-vivid yellow” to “#cc5500-strong orange”); cell lumen (L1 × W1) = 16–52 × 14–33 µm, episporium (L2 × W2) 22–59 × 20–39 µm, including exosporium (L3 × W3) 29–70 × 26–54, spore wall thickness (episporium) 1.0–5.0 µm. Ascospores (L2 × W2) in 1-spored asci 46–60 × 32–40 µm, 2-spored asci 31–48 × 24–33 µm, 3-spored asci 24–44 × 22–33 µm, 4-spored asci 27–39 × 18–28 µm, 5-spored asci 28–34 × 22–27 µm. Exosporium consisting of a reticulum of complete polygonal alveolae with 4–6 sides, 2.0–7.0 µm high, 2.0–10 µm wide; meshes 3–6 and 2–3 across spore length and width, respectively.

Habitat: it seems to be a species associated with thermophilic oak species *Q. pubescens*, *Q. ilex* L. [26], and *Quercus coccifera* L., in central-south Italy and Greece.

Additional specimens examined: **Greece**: Attica, Acharnes, Mt. Parnitha, *Q. coccifera* and *Pinus halepensis* Mill., 12 July 2013, leg. P. Kladopoulou, det. G. Pacioni (AQUI 10229); **Italy:** Abruzzo, Scoppito (AQ), *Q. pubescens*, 1 February 1988, leg. C. Visca, det. G. Pacioni (AQUI 4882); L’Aquila (AQ), Colle di Sassa, *Q. pubescens*, 12 December 2012, leg. D. Marinucci, det. G. Pacioni (AQUI 8761, paratype); L’Aquila (AQ), *Q. pubescens*, 27 November 2011, leg. D. Marinucci, det. G. Pacioni (AQUI 7165); L’Aquila (AQ), *Q. pubescens*, 2 January 2012, leg D. Marinucci, det. G. Pacioni (AQUI 8514); L’Aquila (AQ), *Q. pubescens*, 3 December 2012, leg. D. Marinucci, det. G. Pacioni (AQUI 10323 paratype); central Apennines without indications (AQUI 292, AQUI 10240).

The ITS and EF1-α sequences of this species fall into clade III after Benucci et al. [25] and into subclade III after Marozzi et al. [26], respectively.

## 6. Conclusions

This study clarifies the taxonomy of a commercially important truffle group, and will be useful to support further studies on ecology, cultivation, and foodomics of these species. The specific primer pairs tested in this study can be used for rapid and easy identification of members of the three studied species. Finally, specific investigations on VOC composition are needed to define the aroma repertoires existing in the *T. mesentericum* complex and to commercially promote these species.

## Figures and Tables

**Figure 1 jof-07-01090-f001:**
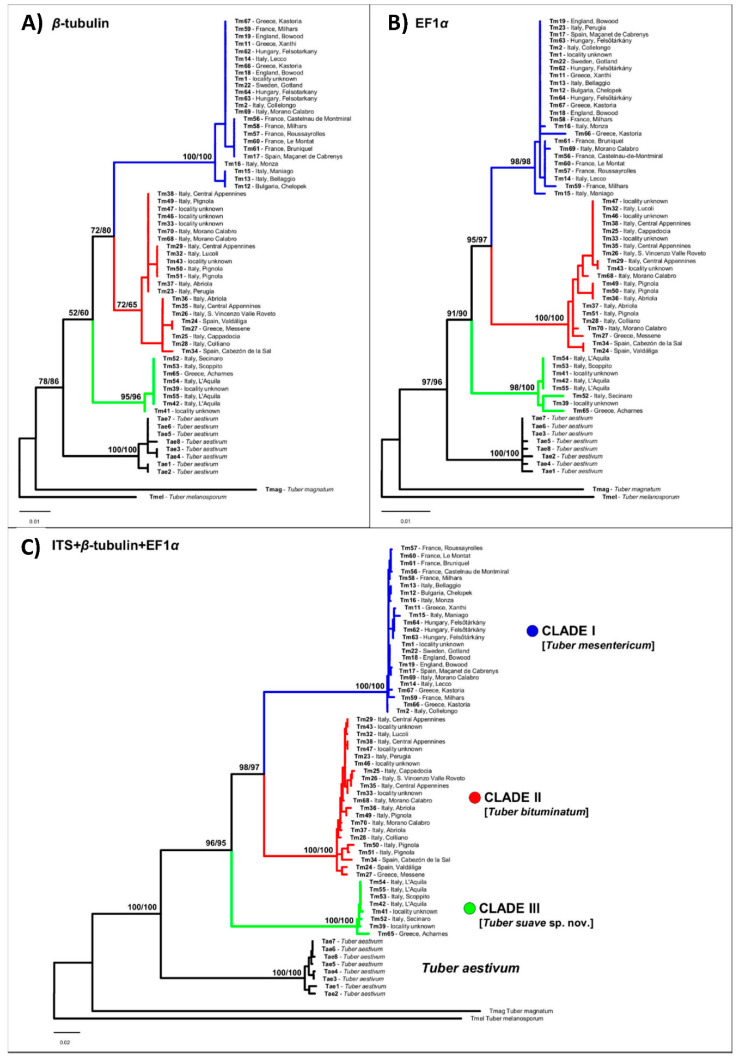
Phylogeny of *T. mesentericum* complex based on the single-loci datasets of β-tubulin (**A**) and elongation factor 1-α (**B**) and the concatenated dataset ITS + β-tub + EF-1-α (**C**). Bootstrap (BS) and rapid bootstrap (rBS) values (>70) are reported on nodes. Specimens are labelled by their geographic origin and ID, listed in Table 1 and Appendix A.

**Figure 2 jof-07-01090-f002:**
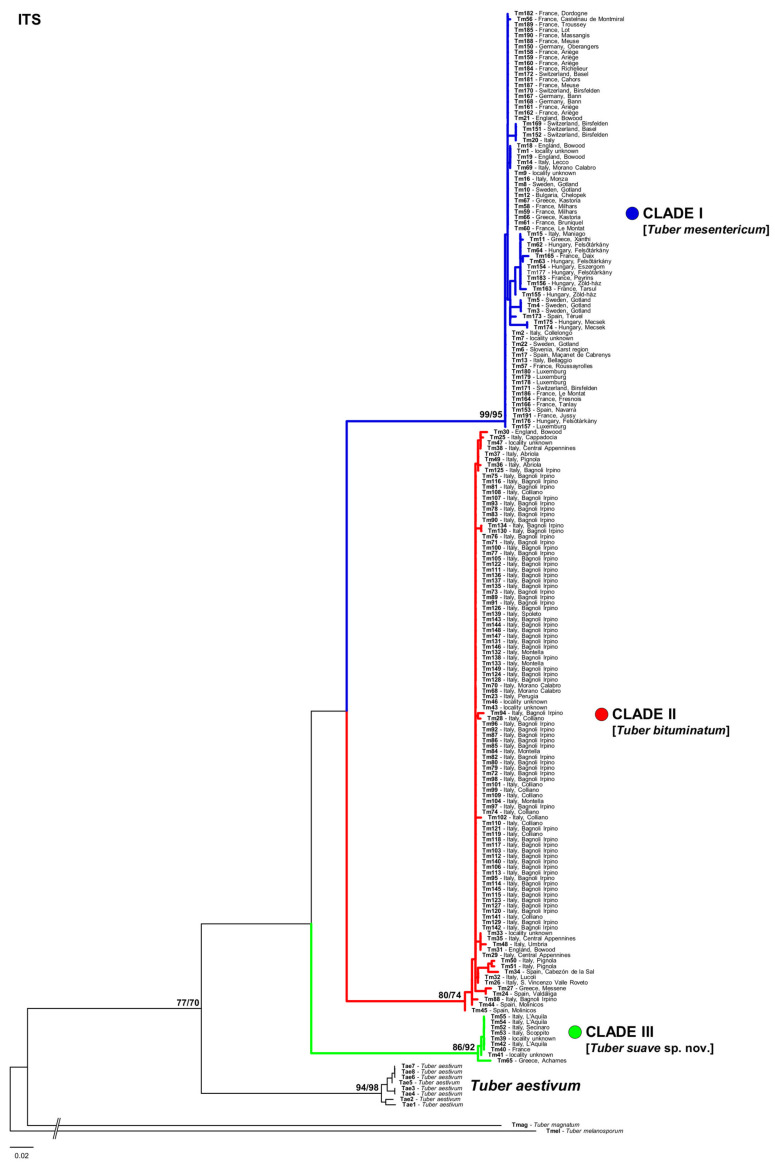
Phylogeny of *T. mesentericum* complex, based on the ITS rDNA dataset. Bootstrap (BS) and rapid bootstrap (rBS) values (>70) are reported on nodes. Specimens are labelled by their geographic origin and ID, listed in Table 1 and Appendix A.

**Figure 3 jof-07-01090-f003:**
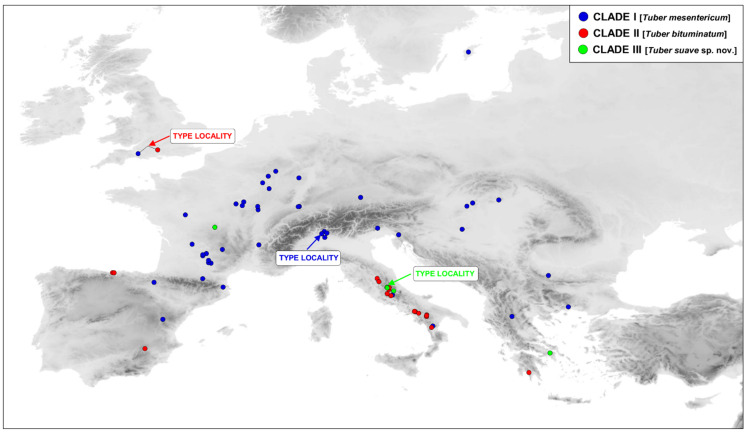
Geographic distribution of main clades, based on specimens analyzed in this study. For each clade, the corresponding taxon and type locality are indicated.

**Figure 4 jof-07-01090-f004:**
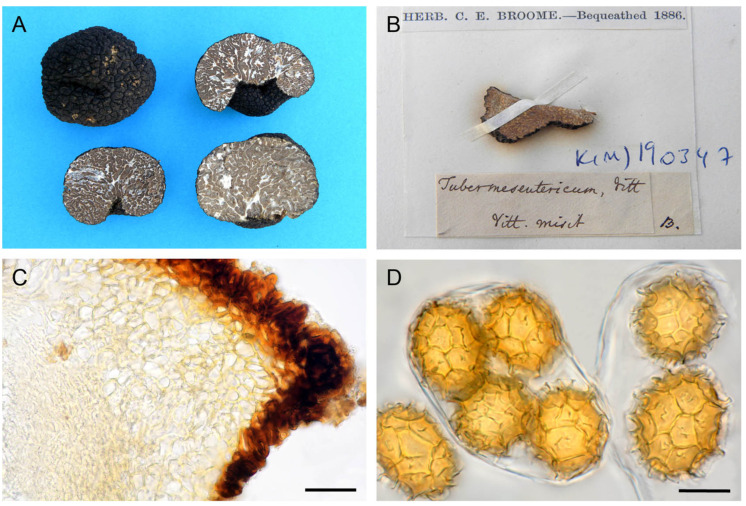
*T. mesentericum* s.s. (**A**) Ascomata in section (neotype). (**B**) Fragment of the ascoma K(M)190347 from the Kew herbarium identified as *T. mesentericum* s.s. (**C**) Peridium structure. (**D**) Asci and ascospores. Bars = 20 µm.

**Figure 5 jof-07-01090-f005:**
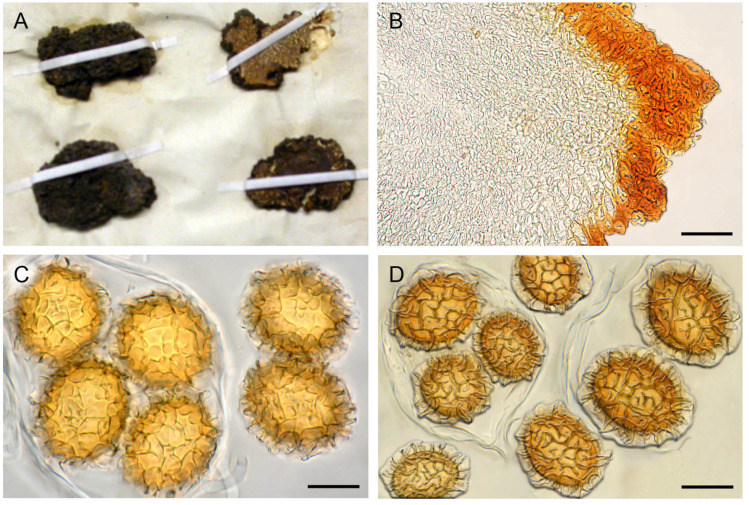
*T. bituminatum*. (**A**) Ascomata in section (lectotypes). (**B**) Peridium structure. (**C**) Asci and ascospores with regular reticula. (**D**) Asci and ascospores with labyrinthine pseudoreticula. Bars = 20 µm.

**Figure 6 jof-07-01090-f006:**
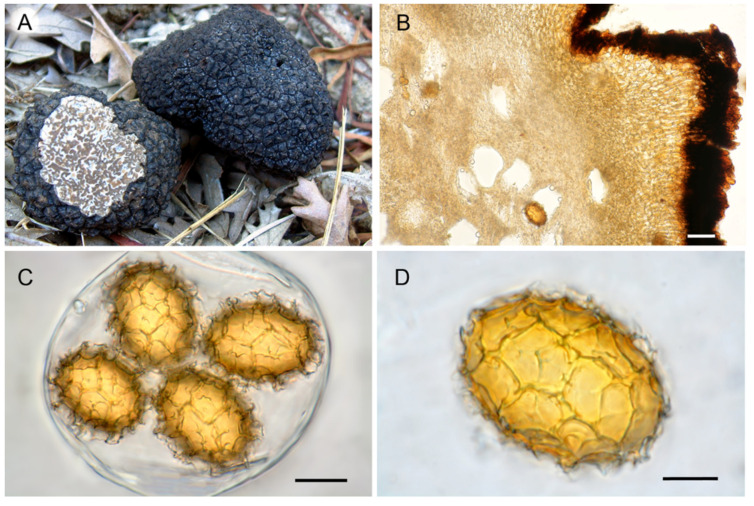
*T. suave* sp. nov. (**A**) Ascoma in section (holotype). (**B**) Peridium structure. (**C**) Asci and ascospores. (**D**) Ascospore. Bars = 50 µm (**B**), 20 µm (**C**), 10 µm (**D**).

**Table 1 jof-07-01090-t001:** Accession numbers and collection locality of the specimens analyzed in this study.

ID	Clade	Locality	ITS	β-tub	EF-1-α
Tm2	I	Italy—Collelongo	KP686246	KT456266	OL753348
Tm11	I	Greece—Xanthi	OL711588	OL753394	OL753349
Tm12	I	Bulgaria—Chelopek	OL711589	OL753395	OL753350
Tm13	I	Italy—Bellagio	OL711590	OL753396	OL753351
Tm14	I	Italy—Lecco	OL711591	OL753397	OL753352
Tm15	I	Italy—Maniago	OL711592	OL753398	OL753353
Tm16	I	Italy—Monza	OL711593	OL753399	OL753354
Tm17	I	Spain—Maçanet de Cabrenys	OL711594	OL753400	OL753355
Tm18	I	England—Bowood	OL711595	OL753401	OL753356
Tm19	I	England—Bowood	OL711596	OL753402	OL753357
Tm20	I	Italy	KT456267	-	-
Tm21	I	England—Bowood	OL711597	-	-
Tm22	I	Sweden—Gotland	AJ888043	KT456265	JX022595
Tm56	I	France—Castelnau-de-Montmiral	OL711598	OL753418	OL753379
Tm57	I	France—Roussayrolles	OL711599	OL753419	OL753380
Tm58	I	France—Milhars	OL711600	OL753420	OL753381
Tm59	I	France—Milhars	OL711601	OL753421	OL753382
Tm60	I	France—Le Montat	OL711602	OL753422	OL753383
Tm61	I	France—Bruniquel	OL711603	OL753423	OL753384
Tm62	I	Hungary—Felsőtárkány	OL711604	OL753424	OL753385
Tm63	I	Hungary—Felsőtárkány	OL711605	OL753425	OL753386
Tm64	I	Hungary—Felsőtárkány	OL711606	OL753426	OL753387
Tm66	I	Greece—Kastoria	OL711607	OL753428	OL753389
Tm67	I	Greece—Kastoria	OL711608	OL753429	OL753390
Tm69	I	Italy—Morano Calabro	OL711609	OL753431	OL753392
Tm23	II	Italy—Perugia	KP686243	KT456263	OL753358
Tm24	II	Spain—Valdáliga	OL711610	OL753403	OL753359
Tm25	II	Italy—Cappadocia	OL711611	OL753404	OL753360
Tm26	II	Italy—S. Vincenzo Valle Roveto	OL711612	OL753405	OL753361
Tm27	II	Greece—Messene	OL711613	OL753406	OL753362
Tm28	II	Italy—Colliano	OL711614	OL753407	OL753363
Tm29	II	Italy—Central Appennines	KP686240	KT456261	OL753364
Tm30	II	England—Bowood	OL711615	-	-
Tm31	II	England—Bowood	OL711616	-	-
Tm32	II	Italy—Lucoli	KP686239	KT456260	OL753365
Tm34	II	Spain—Cabezón de la Sal	OL711617	OL753408	OL753366
Tm35	II	Italy—Central Appennines	KP686242	KX290123	OL753367
Tm36	II	Italy—Abriola	KP686245	KX290124	OL753368
Tm37	II	Italy—Abriola	KP686244	OL753409	OL753369
Tm38	II	Italy—Central Appennines	KP686241	KT456262	OL753370
Tm68	II	Italy—Morano Calabro	OL711618	OL753430	OL753391
Tm49	II	Italy—Pignola	OL711619	OL753411	OL753372
Tm50	II	Italy—Pignola	OL711620	OL753412	OL753373
Tm51	II	Italy—Pignola	OL711621	OL753413	OL753374
Tm70	II	Italy—Morano Calabro	OL711622	OL753432	OL753393
Tm52	III	Italy—Secinaro	OL711623	OL753414	OL753375
Tm53	III	Italy—Scoppito	OL711624	OL753414	OL753376
Tm54	III	Italy—L’Aquila	OL711625	OL753416	OL753377
Tm55	III	Italy—L’Aquila	OL711626	OL753417	OL753378
Tm65	III	Greece—Acharnes	OL711627	OL753427	OL753388
Tm42	III	Italy—L’Aquila	OL711628	OL753410	OL753371

**Table 2 jof-07-01090-t002:** Sequences of the clade-specific primer pairs designed in this study.

Clade	Forward (5′→3′)	Reverse (5′→3′)	Amplicon Size (bp)
I	ATAACGAGAAGTCTGAACC	CTGCTCTACGCTTATCACA	357
II	CGTGAACACACTTTGGACA	CTGCCTTACGCTAATTACAAC	372
III	ACTTGGTAAACTGAAGCAGAC	CTTCACGCCGATTACAACG	365

**Table 3 jof-07-01090-t003:** Sporal dimensions of ascomata belonging to the three clades inferred in this study.

	Clade I		Clade II		Clade III	
	Min–Max	Mean ± SD	Min–Max	Mean ± SD	Min–Max	Mean ± SD
L1	18–53	27.84 ± 6.64	16–52	27.41 ± 5.98	16–52	27.64 ± 6.28
L2	24–61	33.15 ± 7.12	22–61	33.06 ± 4.66	22–59	33.47 ± 5.91
L3	30–70	41.71 ± 8.06	29–70	43.71 ± 7.24	29–70	43.24 ± 7.24
W1	16–31	20.99 ± 3.49	14–33	21.41 ± 3.69	14–33	21.49 ± 3.69
W2	20–41	26.84 ± 4.38	20–43	25.04 ± 3.63	20–39	27.11 ± 3.78
W3	26–56	35.84 ± 5.61	26–54	37.13 ± 5.62	26–54	36.85 ± 5.62
Q1 (L1/W1)	1.00–1.83	1.32 ± 0.19	1.00–2.21	1.28 ± 0.19	1.00–1.93	1.38 ± 0.16
Q2 (L2/W2)	1.00–1.69	1.23 ± 0.14	1.00–2.09	1.23 ± 0.15	1.00–1.75	1.31 ± 0.16
Q3 (L3/W3)	0.92–1.45	1.17 ± 0.12	0.95–1.67	1.17 ± 0.12	1.00–1.54	1.22 ± 0.12
L2–L1	3–10	5.43 ± 1.41	3–12	8.86 ± 2.41	2.0–10.0	4.98 ± 1.42
L3–L2	4–16	8.57 ± 2.33	4–16	11.00 ± 2.31	4.0–14.0	8.02 ± 1.98
W2–W1	2–8	5.85 ± 1.41	4–13	9.71 ± 1.80	3.0–10.0	5.03 ± 1.32
W3–W2	5–15	8.74 ± 2.15	9–19	10.14 ± 1.46	4.0–14.0	8.44 ± 2.06
Episp. Thickness ^1^	1.5–5.0	3.14 ± 0.95	1.5–6.5	2.93 ± 0.90	1.0–5.0	2.50 ± 0.68
Exosp. Height ^2^	2.0–8.0	4.07 ± 1.13	2.0–9.5	4.95 ± 1.31	2.0–7.0	4.11 ± 1.01

^1^ Episporium thickness parameters were obtained by halving L2–L1 and W2–W1 values. ^2^ Exosporium height parameters were obtained by halving L3–L2 and W3–W2 values.

## Data Availability

All data obtained and analyzed in this study have been included in this article and its Appendix A.

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
