# Peer review of "Multilocus Phylogeography of the Tuber mesentericum Complex Unearths Three Highly Divergent Cryptic Species"

_jof, 2021, doi:10.3390/jof7121090_

Round 1

Reviewer 1 Report

Dear authors,

Here is my comments, please check it!

Based on multilocus data and including specimens from all over Europe, this paper provides useful data concerning classification of Tuber mesentericum species complex. Phylogenetic analyses consistently recovered three reciprocally monophyletic and well supported clades: Tuber mesentericum, Tuber bituminatum and Tuber suave sp. nov.. In addition, this paper also designed new primer pairs specific for a rapid and easy identification of members of the three studied clades. In my opinion, the manuscript can be accepted for publication after some minor modifications. My comments are as follows: In line 70, “T. mesentericum s.s” should be “T. mesentericum sensu stricto”, because this is the first time the “sensu stricto” has appeared and should not be abbreviation. In line 263, please provide what kind of clade II host is it? In line 267, 268, and 269, please combine phylogenetic and phylogeographic studies in this paper to explore the genetic divergence of interspecies and intraspecies within Tuber.

Author Response

Reviewer 1

In line 70, “T. mesentericum s.s” should be “T. mesentericum sensu stricto”, because this is the first time the “sensu stricto” has appeared and should not be abbreviation.

Done

In line 263, please provide what kind of clade II host is it?

The sentence was modified as follows: “….and their members shows a wide host preference as for the members of clade I (hardwoods and conifers)”.

In line 267, 268, and 269, please combine phylogenetic and phylogeographic studies in this paper to explore the genetic divergence of interspecies and intraspecies within Tuber.

We added the values of ITS divergences within each clade in result section: “Intra-clade divergence at the ITS region was 0.0047 for clade I, 0.0074 for clade II and 0.0032 for clade III using either P-distance or Kimura-2-parameter estimates.” and modified the sentences in discussion as follows: “The clades I and II seem to be sister species that have diverged after the clade III. While the average intra-clade divergence at the ITS marker was < 0.8%, the divergence between these clades was as high as 10% or 13%, values that are well above those observed among the currently accepted Tuber species. For example, inter-specific divergence observed between closely related pairs of currently recognized species belonging to Rufum and Melanosporum clades are in the range 4.7‒7.4% [39]. Bonito et al. [42] suggested a threshold for Tuber species delimitation based on inter-specific divergence ≥ 4%. However, inter-specific divergence below 4% have been reported between closely related species of different clades such as the Excavatum [12], Puberulum [9] and Gibbosum [43] clades.”

Reviewer 2 Report

First of all, I would like to congratulate the authors on their careful and thorough work.

The taxonomy of several large species groups of true truffles is still disordered (e.g. Tuber excavatum, Tuber rufum). This is why it is gratifying to meet the taxonomic processing of the Tuber mesentericum species group in this MS.

A sequencing strategy for shorter ITS sections, but also for the most informative ITS domains, is extremely ingenious. The design and subsequent successful use of clade-specific primers is the methodological innovation that primarily determines the success of this work, as it has also made it possible to evaluate older herbarium samples despite their high degree of DNA degradation. This methodological solution could pave the way for similar works.

Since unique morphological features can be found in all 3 clades, the reviewer thinks of taxa as more pseudocriptic species, but accepts the authors' more cautious approach to evaluating and classifying taxa as cryptic species.

The number of formal, editorial and citation errors is extremely small, and my suggestions for them are given below:

It is recommended to insert an abbreviation chapter after the headings chapter (ITS, VOCs, R :, H :, etc.).

Letter / character and other errors: ad 49, 185, 361/362, 403

Not correct citation: ad 108-109

Rules of the Botanical Nomenclature not complied with (italics, etc.): ad 112, 130, 343, 420, 487, 620, 634, 674

Instead of “pyramidal warts”: (very) flat pyramidal warts are recommended / ad 351, 427, 493, 499

Ad 399 instead of “leg. K. Imre ”: leg. I. Kiss

Ad 354 "with a faint solvent component (trichlorethylene)" - no citation !?

Ad 183 Recommended for use:  Sporal dimensions

Ad 275/277 .. between clades…. (or amongst them?)

Ad 529 Incorrect citation.

Author Response

Thanks to the Reviewer 2 for corrections

Since unique morphological features can be found in all 3 clades, the reviewer thinks of taxa as more pseudocriptic species, but accepts the authors' more cautious approach to evaluating and classifying taxa as cryptic species.

We prefer to be cautious and keep the name ‘cryptic’ because while some subtle differences in morphology of spore or peridium were detected there is a wide overlap of values between species for these variables and there is not any single diagnostic character.

It is recommended to insert an abbreviation chapter after the headings chapter (ITS, VOCs, R, H, etc.).

The journal does not provide an abbreviation chapter; however, we reported in the text the spelling of these abbreviations.

Letter / character and other errors: ad 49, 185, 361/362, 403

At ln 49 the authority “Picco” is correct (see for reference Pacioni et al. 2019 on Italian Journal of Mycology)

At ln 187, we changed “(L2 – L1) and (W2 – W1)” with “(L3 – L2) and (W3 – W2)”

Not correct citation: ad 108-109

We changed citation [28] with [30]

Rules of the Botanical Nomenclature not complied with (italics, etc.): ad 112, 130, 343, 420, 487, 620, 634, 674

All the names were italicized; sorry but the copy and paste activities on the journal template sometimes modified the text format

Instead of “pyramidal warts”: (very) flat pyramidal warts are recommended / ad 351, 427, 493, 499

We added “flat” before “pyramidal warts” everywhere

Ad 399 instead of “leg. K. Imre”: leg. I. Kiss

done

Ad 354 "with a faint solvent component (trichlorethylene)" - no citation!?

There is not a bibliographic reference in this case because it is only a description of the sensorial perception generated by the fresh ascomata of T. mesentericum s.s.; we changed the text as follows: “…with a faint solvent (similar to trichloroethylene)…”

Ad 183 Recommended for use: Sporal dimensions

Changed

Ad 275/277 between clades…. (or amongst them?)

We changed “between the” with “among”

Ad 529 Incorrect citation.

“Marozzi et al. 2020” was replaced with “[26]”